# Variational Anisotropic Gradient-Domain Image Processing

**DOI:** 10.3390/jimaging7100196

**Published:** 2021-09-29

**Authors:** Ivar Farup

**Affiliations:** Department of Computer Science, Norwegian University of Science and Technology (NTNU), 2802 Gjøvik, Norway; ivar.farup@ntnu.no

**Keywords:** variational methods, anisotropic diffusion, gradient-domain image processing, local contrast enhancement

## Abstract

Gradient-domain image processing is a technique where, instead of operating directly on the image pixel values, the gradient of the image is computed and processed. The resulting image is obtained by reintegrating the processed gradient. This is normally done by solving the Poisson equation, most often by means of a finite difference implementation of the gradient descent method. However, this technique in some cases lead to severe haloing artefacts in the resulting image. To deal with this, local or anisotropic diffusion has been added as an ad hoc modification of the Poisson equation. In this paper, we show that a version of anisotropic gradient-domain image processing can result from a more general variational formulation through the minimisation of a functional formulated in terms of the eigenvalues of the structure tensor of the differences between the processed gradient and the gradient of the original image. Example applications of linear and nonlinear local contrast enhancement and colour image Daltonisation illustrate the behaviour of the method.

## 1. Introduction and Background

In 2002, Fattal et al. [1] introduced the method of gradient-domain high-dynamic-range compression. The technique consisted of first computing the gradient field $\nabla u_{0}$ of a high-dynamic-range image $u_{0}:\Omega \to C$, where $\Omega \subset R^{2}$ is the image domain and $C\subset R^{3}$ is the colour space. The gradient was then rescaled nonlinearly as $G=f(\nabla u_{0})$. The resulting tensor field G, which is no longer necessarily a gradient field, was then reintegrated by solving the Poisson equation
(1)$$\nabla^{2}u=\nabla \cdot G$$
to obtain the compressed image. This is often solved by a gradient descent,
(2)$$\frac{\partial u}{\partial t}=\nabla^{2}u-\nabla \cdot G$$

The Poisson equations can be obtained through a variational approach by minimising the functional
(3)$$E\left(u\right)=\frac{1}{2}\int_{\Omega }||\nabla u-{G||}_{F}^{2}\;d\Omega =\int_{\Omega }L\left(u,\nabla u\right)\;d\Omega$$
where $\Omega \subset R^{2}$ is the image domain and the index *F* indicates the Frobenius norm taken over both image and colour coordinates. This is done by solving the corresponding Euler–Lagrange equations,
(4)$$\nabla \cdot \left(\frac{\partial L}{\partial \nabla u}\right)-\frac{\partial L}{\partial u}=0$$
leading to Equation (Equation 1).

Perez et al. [2] soon generalised this into the technique called Poisson Image Editing, and showed that it allowed for a broad range of image processing applications such as image inpainting, seamless cloning, texture transfer, feature exchange, insertion of transparent objects through gradient mixing, texture flattening, local illumination correction, and seamless tiling. All this is obtained by various processing of the gradient or gradients of the original images to obtain the tensor field G used in Equation (Equation 1).

One significant challenge of Poisson Image Editing is that it tends to produce visual haloing or blurring artefacts in the reconstructed images. The case of G=0 in Equation (Equation 1) is often used for image denoising. For greyscale images, in order to stop the diffusion at edges, and thus reduce the blurring, Perona and Malik [3] introduced an image-dependent, local, nonlinear diffusion method described by the equation
(5)$$\frac{\partial u}{\partial t}=\nabla \cdot \left(D\left(s\right)\nabla u\right)$$
where $s={\left|\nabla u\right|}^{2}$ describing the image ‘structure’ has been introduced. (Despite the fact that the method is isotropic, but nonlinear and local, the authors termed it ‘anisotropic diffusion’ in the original publication [3]. This has caused considerable confusion in the terminology in the following literature.) They proposed two different diffusion coefficients with different properties,
(6)$$\begin{array}{cc} D(s) & =\exp\left(-\frac{s}{K^{2}}\right) \end{array}$$
(7)$$\begin{array}{cc} D(s) & =\frac{1}{1+s/K^{2}} \end{array}$$

Instead of designing partial differential equations (PDEs) directly, Rudin et al. [4] used a variational approach like Equation (Equation 3) and introduced the concept of total variation. They showed that the minimisation of the functional
(8)$$E=\int_{\Omega }\left|\nabla u\right|\;d\Omega$$
leads to the PDE
(9)$$\frac{\partial u}{\partial t}=\nabla \cdot \left(\frac{\nabla u}{\left|\nabla u\right|}\right)$$

Comparing with the Perona–Malik diffusion, Equation (Equation 5), we see that it can be written in the same form choosing
(10)$$D\left(s\right)=\frac{1}{\sqrt{s}}$$

For total variation and Perona–Malik diffusion, the extension to colour images is not that straight forward. Blomgren and Chan [5] introduced the concept of colour total variation by using a functional as an $\ell_{2}$ norm of the functionals in Equation (Equation 8) for each of the colour channels resulting in
(11)$$E=\sqrt{\sum_{i}E_{i}^{2}}$$
where $E_{i}$ is the total variation for each colour channel according to Equation (Equation 8).

Later approaches have been based on the structure tensor by Di Zenzo [6] and Bigun and Granlund [7], with components
(12)S=∇u·∇u
where the dot product is taken over the colour coordinates. Sapiro and Ringach [8] proposed to use the eigenvalues
(13)$$\lambda_{\pm }=\frac{1}{2}\left(S_{11}+S_{22}\pm \sqrt{{\left(S_{11}-S_{22}\right)}^{2}+4S_{12}^{2}}\right)$$
and the corresponding eigenvectors $\theta_{\pm }$ of the structure tensor as a basis for constructing the diffusion equations. In terms of these eigenvalues, an alternative to the colour total variation by Blomgren and Chan [5] can be obtained by the functional
(14)$$E=\int_{\Omega }\sqrt{\lambda_{+}+\lambda_{-}}\;d\Omega =\int_{\Omega }\sqrt{s}\;d\Omega$$

For greyscale images, where $\lambda_{+}={\left|\nabla u\right|}^{2}$ and $\lambda_{-}=0$, this reduces to total variation. For colour images, the corresponding Euler–Lagrange equations become
(15)$$\frac{\partial u}{\partial t}=\nabla \cdot \left(\frac{\nabla u}{{\left||\nabla u|\right|}_{F}}\right)$$
which again is on the form of Equation (Equation 5) with $D\left(s\right)=1/\sqrt{s}=1/\sqrt{\lambda_{+}+\lambda_{-}}=1/\sqrt{S_{11}+S_{22}}=1/{\left||\nabla u|\right|}_{F}$. Notice the coupling between the colour channels introduced by the sum in the denominator of Equation (Equation 15).

Tschumperlé and Deriche [9] extended this approach to a anisotropic diffusion by introducing the general Lagrangian density $\psi (\lambda_{+},\lambda_{-})$ in the functional
(16)$$E=\int_{\Omega }\psi \left(\lambda_{+},\lambda_{-}\right)\;d\Omega$$

The corresponding Euler–Lagrange equations are
(17)$$\frac{\partial u}{\partial t}=\nabla \cdot \left(D\cdot \nabla u\right)$$
where D is the diffusion tensor
(18)$$D=2\left(\frac{\partial \psi }{\partial \lambda_{+}}\theta_{+}\theta_{+}^{T}+\frac{\partial \psi }{\partial \lambda_{-}}\theta_{-}\theta_{-}^{T}\right)$$
where $\theta_{\pm }$ are the eigenvectors of the structure sensor S. It should be noted that this actually encompasses all previously presented diffusion methods as follows:(19)$$\psi \left(\lambda_{+},\lambda_{-}\right)=\sqrt{\lambda_{+}+\lambda_{-}}$$
gives the solution of Sapiro and Ringach [8] for colour images and total variation of Rudin et al. [4] for greyscale images,
(20)$$\begin{array}{cc} \psi (\lambda_{+},\lambda_{-}) & =-K^{2}\exp\left(-s/K^{2}\right)\;\mathrm{and} \end{array}$$
(21)$$\begin{array}{cc} \psi (\lambda_{+},\lambda_{-}) & =K^{2}\ln\left(1+s/K^{2}\right) \end{array}$$
give the two equations of Perona and Malik [3], and
(22)$$\psi (\lambda_{+},\lambda_{-})=s/2$$
gives the classical linear diffusion, Equation (Equation 2).

Choosing
(23)$$\psi \left(\lambda_{+},\lambda_{-}\right)=\phi \left(\lambda_{+}+\lambda_{-}\right)=\phi \left(s\right)$$
in general leads to isotropic equations, since then $\partial \psi /\partial \lambda_{+}=\partial \psi /\partial \lambda_{-}=\phi^{\prime }\left(s\right)$ and the diffusion tensor in Equation (Equation 18) reduces to the scalar diffusion coefficient $D=2\phi^{\prime }\left(s\right)$. With
(24)$$\psi \left(\lambda_{+},\lambda_{-}\right)=\phi^{+}\left(\lambda_{+}\right)+\phi^{-}\left(\lambda_{-}\right),$$
the diffusion in the mutually orthogonal directions of maximal and minimal change can be controlled independently, like used by, e.g., Farup [10].

For other application of Poission image editing, i.e., applications of Equation (Equation 2) where G≠0, extensions to anisotropic and edge-preserving methods have been obtained by ad hoc modifications of Equation (Equation 2). This has been done for e.g., colour gamut mapping [11], colour image demosaicing [12], colour-to-greyscale conversion [13], and colour image Daltonisation [10]. Common to these is that they solve an equation on the form
(25)$$\frac{\partial u}{\partial t}=\nabla \cdot \left(D\cdot (\nabla u-G)\right)$$
where D is a diffusion tensor constructed from the structure tensor like, e.g., Equation (Equation 18).

However, a unifying variational formulation of anisotropic gradient-domain image processing has not yet been given. In this paper, we provide this by introducing the difference structure tensor based on the difference of the original image gradient ∇u and the tensor field G and follow the process of Tschumperlé and Deriche [9]. The resulting PDE is similar, but not identical, to the ones obtained by the ad hoc modification of Poisson Image editing, Equation (Equation 25). For the applications to linear and nonlinear local contrast enhancement and colour image Daltonisation, we show that the two approaches gives very similar results.

## 2. Variational Anisotropic Gradient-Domain Formulation

The variational formulation of anisotropic diffusion by Tschumperlé and Deriche [9] was based on the structure tensor S, Equation (Equation 18). In analogy, we introduce the difference structure tensor,
(26)$$S^{\prime }=\left(\nabla u-G\right)\cdot \left(\nabla u-G\right)$$
with eigenvalues
(27)$$\lambda_{\pm }^{\prime }=\frac{1}{2}\left(S_{11}^{\prime }+S_{22}^{\prime }\pm \sqrt{{\left(S_{11}^{\prime }-S_{22}^{\prime }\right)}^{2}+4{S_{12}^{\prime }}^{2}}\right)$$
and corresponding eigenvectors $\theta_{\pm }^{\prime }$.

In analogy with Tschumperlé and Deriche [9], we define the functional in terms of the eigenvalues of this difference structure tensor,
(28)$$E=\int_{\Omega }\psi \left(\lambda_{+}^{\prime },\lambda_{-}^{\prime }\right)\;d\Omega$$

The corresponding Euler–Lagrange equations are
(29)$$\nabla \cdot \left(\frac{\partial \psi }{\partial \nabla u}\right)-\frac{\partial \psi }{\partial u}=0$$

It is clear that ∂ψ/∂u=0. To derive the explicit form of the Euler–Lagrange equations, we will need to look at the individual components. Using Greek indices for the colour coordinates, Latin indices for the spatial coordinates, letting a comma denote partial differentiation with respect to the following coordinates, and using Einstein’s convention of summing over repeated indices, the expression within the parenthesis in the first term can be written
(30)$$\frac{\partial \psi }{\partial u_{,i}^{\rho }}=\frac{\partial \psi }{\partial \lambda_{p}^{\prime }}\frac{\partial \lambda_{p}^{\prime }}{\partial S_{kl}^{\prime }}\frac{\partial S_{kl}^{\prime }}{\partial u_{,i}^{\rho }}$$
where the index *p* is used for summing over the two eigenvalues of the difference structure tensor. The first factor, $\partial \psi /\partial \lambda_{p}^{\prime }$, can be computed directly since the Lagrangian ψ is defined explicitly in terms of the eigenvalues of the difference structure tensor, and will thus depend on the design of the Lagrangian density.

The second factor can be computed implicitly following the method of Tschumperlé and Deriche [9] as follows:(31)$$\begin{array}{cc} \delta_{i}^{k}\delta_{j}^{l} & =\frac{\partial S_{kl}^{\prime }}{\partial S_{ij}^{\prime }}\\ \; & =\frac{\partial \lambda_{p}^{\prime }}{\partial S_{ij}^{\prime }}\theta_{pk}^{\prime }\theta_{pl}^{\prime }+\lambda_{p}^{\prime }\frac{\partial \theta_{pk}^{\prime }}{\partial S_{ij}^{\prime }}\theta_{pl}^{\prime }+\lambda_{p}^{\prime }\theta_{pk}^{\prime }\frac{\partial \theta_{pl}^{\prime }}{\partial S_{ij}^{\prime }} \end{array}$$

Multiplying with $\theta_{mk}^{\prime }$ and $\theta_{ml}^{\prime }$, summing over the *k* and *l* indexes, using that $\theta_{mk}^{\prime }\theta_{pk}^{\prime }=\delta_{pm}$, and $\theta_{pk}^{\prime }\left(\partial \theta_{pk}^{\prime }/\partial S_{ij}^{\prime }\right)=0$, the latter due to the orthonormality of the $\theta^{\prime }$s gives
(32)$$\frac{\partial \lambda_{p}}{\partial S_{kl}^{\prime }}=\theta_{pk}^{\prime }\theta_{pl}^{\prime }$$
(no sum).

The last term, $\partial S_{kl}^{\prime }/\partial u_{,i}^{\rho }$ deviates from the derivation of Tschumperlé and Deriche due to the definition of the difference structure tensor, Equation (Equation 26),
(33)$$\frac{\partial S_{kl}^{\prime }}{\partial u_{,i}^{\rho }}=\delta_{ik}\left(u_{,l}^{\rho }-v_{l}^{\rho }\right)+\delta_{il}\left(u_{,k}^{\rho }-v_{k}^{\rho }\right)$$

Inserted into Equation (Equation 30) and exploiting the symmetry of Equation (Equation 32) gives
(34)$$\frac{\partial \psi }{\partial u_{,k}^{\rho }}=2\frac{\partial \psi }{\partial \lambda_{p}^{\prime }}\theta_{pk}^{\prime }\theta_{pl}^{\prime }\left(u_{,k}^{\rho }-v_{k}^{\rho }\right)=D_{kl}^{\prime }\left(u_{,k}^{\rho }-v_{k}^{\rho }\right)$$
where the diffusion tensor
(35)$$D_{kl}^{\prime }=2\frac{\partial \psi }{\partial \lambda_{p}^{\prime }}\theta_{pk}^{\prime }\theta_{pl}^{\prime }$$
has been defined.

Inserting this into Equation (Equation 29) and solving by gradient descent gives the variational anisotropic gradient-domain image processing PDE (switching back to the regular vector notation):(36)$$\frac{\partial u}{\partial t}=\nabla \cdot \left(D^{\prime }\cdot \left(\nabla u-G\right)\right),\;\mathrm{where}\;D^{\prime }=2\left(\frac{\partial \psi }{\partial \lambda_{+}^{\prime }}\theta_{+}^{\prime }{\theta_{+}^{\prime }}^{T}+\frac{\partial \psi }{\partial \lambda_{-}^{\prime }}\theta_{-}^{\prime }{\theta_{-}^{\prime }}^{T}\right)$$

The form of this equation is similar to the one based on the ad hoc approach, Equation (Equation 25). The only difference is that, in this case, the diffusion tensor is computed from the difference structure tensor, and not the common structure tensor.

## 3. Results and Discussion

In order to demonstrate the usefulness of the proposed method, we apply it to three example problems: linear local contrast enhancement, nonlinear local contrast enhancement, and colour image Daltonisation.

### 3.1. Implementation

The algorithm represented by Equation (Equation 36) is implemented using the finite difference method (FDM). For the time derivative, the explicit (forward) Euler method is used. For the spatial derivatives, forward differences are used for the ∇u and G terms, and backward difference is used for the divergence in order to balance the overall resulting numerical scheme. The resulting code is available online. https://github.com/ifarup/variational-anisotropic-gradient-domain (accessed on 28 September 2021).

A selection of colourful and detailed test images was downloaded from Pixnio. https://pixnio.com (accessed on 28 September 2021). The images are shown in Figure 1. All images are available under the CC0 licence, https://creativecommons.org/share-your-work/public-domain/cc0/ (accessed on 28 September 2021).

### 3.2. The Diffusion Tensor

The purpose of the diffusion tensor is to direct the diffusion of the image information. The idea is that the diffusion should be avoided across the edges of the image, but allowed to some degree along the edges. The expression for the actual diffusion tensor is given in Equation (Equation 36). A choice has to be made for the function ψ. Here, we will use the Perona–Malik motivated functional
(37)$$\psi \left(\lambda_{+}^{\prime },\lambda_{-}^{\prime }\right)=K^{2}\ln\left(1+\lambda_{+}^{\prime }/K^{2}\right)+K^{2}\ln\left(1+\lambda_{-}^{\prime }/K^{2}\right)$$

The resulting diffusion coefficients along and across the edges, represented by $2\partial \psi^{\prime }/\partial \lambda_{+}^{\prime }$ and $2\partial \psi^{\prime }/\partial \lambda_{-}^{\prime }$, respectively, are shown for one of the test images in Figure 2, where it is easily seen how the edges influence the directional diffusion.

### 3.3. Linear Local Contrast Enhancement

Local contrast enhancement in the gradient domain is a technique that is known to be particularly prone to haloing problems. First, we follow a simple approach and define the contrast enhancement as a scalar multiplication of the original image gradient
(38)G=a∇u
where a>1.

We compare this with the standard Poisson method, Equation (Equation 2), and the ad hoc addition with the diffusion tensor derived from the structure tensor instead of the difference structure tensor, Equation (Equation 18). We use a=2 in order to obtain a quite extreme contrast enhancement, and use with $K=10^{-3}$ for the variational approach based on the difference structure tensor, and $K=3\times 10^{-4}$ for the ad hoc solution based on the standard structure tensor. Different constants are needed since the scaling of the structure tensor will be different in the two cases. Example results are shown in Figure 3. We can see that, with this choice of *K*, the results of the variational and the ad hoc approaches are indistinguishable, whereas the Poisson solutions exhibit severe haloing artefacts, as expected.

### 3.4. Nonlinear Local Contrast Enhancement

A somewhat more sophisticated contrast enhancement can be achieved by using the nonlinear gamma compression in the gradient domain,
(39)$$G={\mathrm{sign}\left(\nabla u\right)|\nabla u|}^{\gamma }\;\mathrm{with}\;0<\gamma <1$$
computed element-wise. Solving as for the linear case with the same choice of parameters as above gives the results shown in Figure 4. We can observe the indistinguishable behaviour between the variational approach and the ad hoc solution, and that the haloing problem of the Poisson solution is solved.

### 3.5. Colour Image Daltonisation

Colour image Daltonisation denotes the recolouring of colour images to increase detail visibility for colour-deficient observers. It is a problem that is known to be particularly prone to the loss of finer image details and textures. It has previously been shown that this process can be performed in the gradient-domain [10] by constructing a gradient field based on the original image as
(40)$$G=\nabla u_{0}+\left(\nabla u_{0}\cdot e_{d}\right)e_{c}$$
where $u_{0}$ is the original colour image, $e_{d}$ is the first principal component of the difference between the original image and a colour-vision-deficiency simulation the same image, and $e_{c}$ is a unit vector of maximum chromatic visibility for the colour-vision-deficient observer (in practice, a vector orthogonal to both the lightness axis and $e_{d}$). Figure 5 shows the resulting daltonised images (left) and the corresponding colour-vision-deficient simulations (right). We observe that the variational approach solves the same problem of lost detail visibility as the ad hoc anisotropic solution, and again gives results that are indistinguishable.

## 4. Conclusions

In this paper, a variational formulation of anisotropic gradient-domain image processing has been introduced. It generalises previous formulations of anisotropic processing and also introduces the difference structure tensor. The resulting PDE deviates somewhat from the one usually found when anisotropic diffusion is added in and ad hoc manner to do gradient-domain image processing. The difference is that the diffusion tensor is based on the difference structure tensor rather than the conventional structure tensor. The example applications of linear and nonlinear local contrast enhancement and colour image Daltonisation illustrate that the behaviour is very similar to what is obtained using the conventional approach. This indicates that the proposed variational formulation is well suited for rigorous derivations of anisotropic gradient-domain image processing for a broad range of applications.

## Figures and Tables

**Figure 1 jimaging-07-00196-f001:**
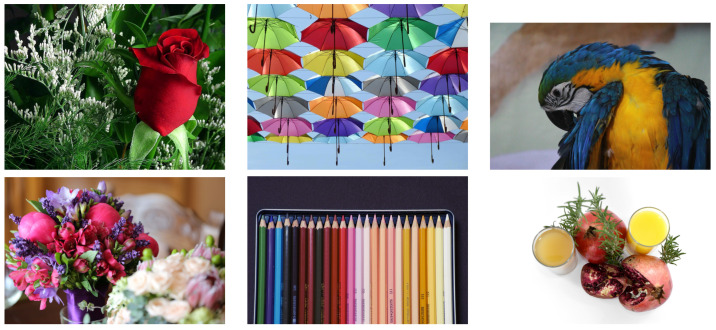
Test images from Pixnio. All images are available under the CC0 licence.

**Figure 2 jimaging-07-00196-f002:**
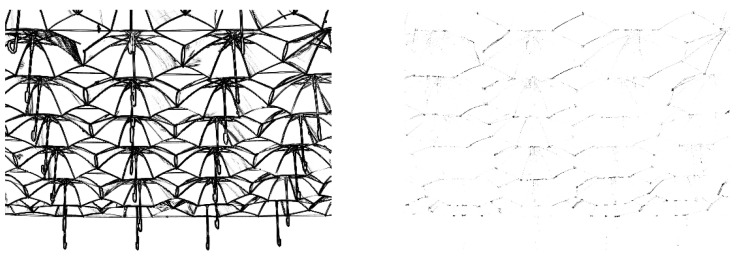
The diffusion coefficients $2\partial \psi^{\prime }/\partial \lambda_{+}^{\prime }$ (**left**) and $2\partial \psi^{\prime }/\partial \lambda_{-}^{\prime }$ (**right**) for one of the test images.

**Figure 3 jimaging-07-00196-f003:**
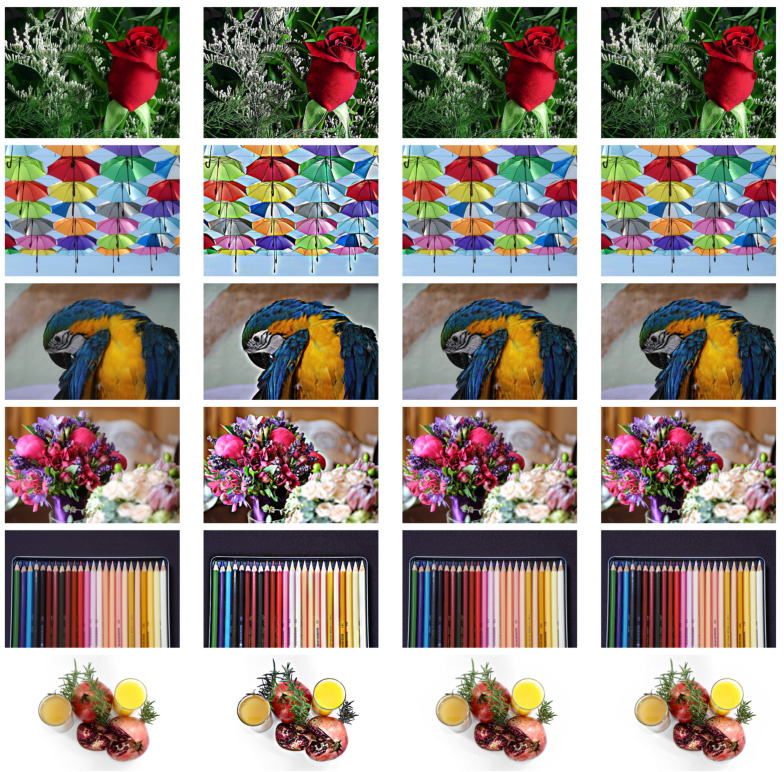
Example results for linear contrast enhancement with a=2. Original images in the first column, Poisson solution in the second, ad hoc anisotropic diffusion with $K=10^{-3}$ in the third, and the proposed variational with $K=3\times 10^{-4}$ in the fourth. All images are available under the CC0 licence.

**Figure 4 jimaging-07-00196-f004:**
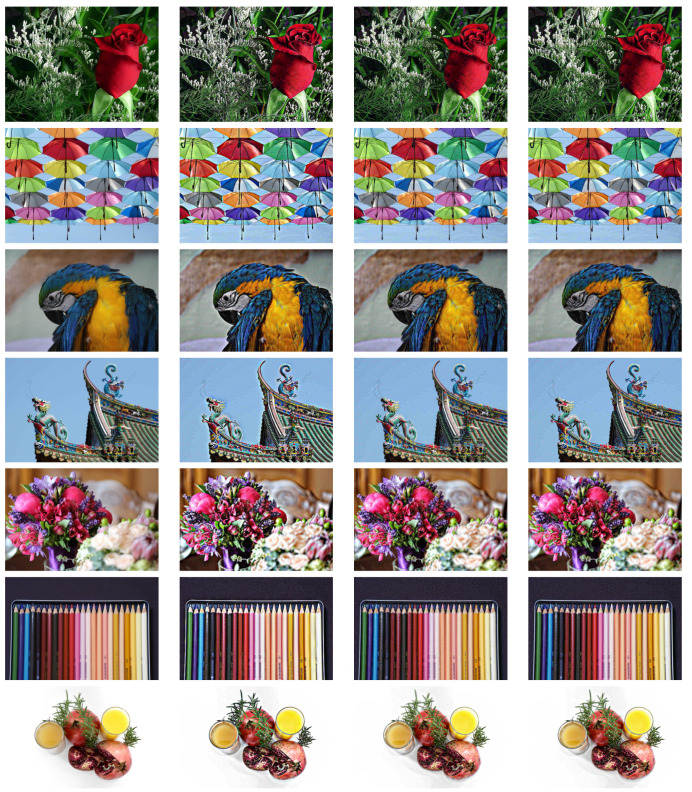
Example results for gamma contrast enhancement with γ=0.7. Original images in the first column, Poisson solution in the second, ad hoc anisotropic diffusion with $K=10^{-3}$ in the third, and the proposed variational with $K=0.3\times 10^{-4}$ in the fourth. All images are available under the CC0 licence.

**Figure 5 jimaging-07-00196-f005:**
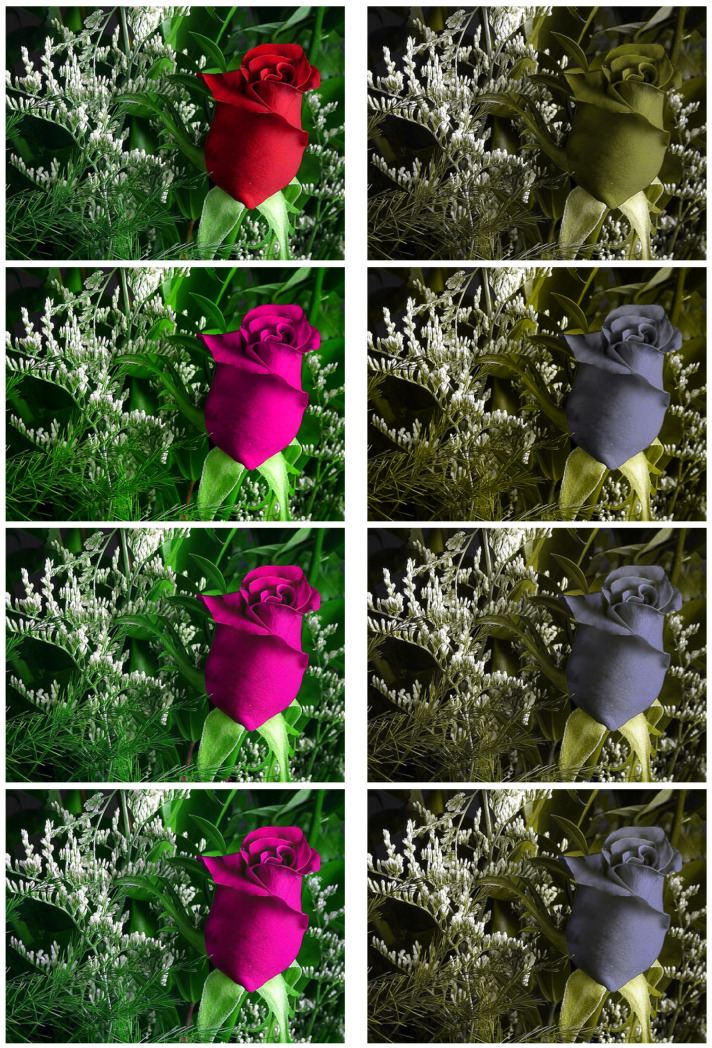
Example of colour image Daltonisation by the proposed method. Colour images in the left column, and corresponding colour–vision–deficiency simulations in the right. Top to bottom: original image, simple global Daltonisation, ad hoc anisotropic gradient domain Daltonisation, and the proposed variational gradient-domain solution.

## Data Availability

The code is available at https://github.com/ifarup/variational-aniso-tropic-gradient-domain (accessed on 28 September 2021).

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
