# Peer review of "Variational Anisotropic Gradient-Domain Image Processing"

_2313-433X, 2021, doi:10.3390/jimaging7100196_

Round 1

Reviewer 1 Report

I like very much the motivation, the proposed idea and the clarity of exposition of previous work.

However, there are some issues that in my view would greatly improve the quality of the paper if they are properly addressed:

1) The notation used is quite confusing because it's not the standard notation for the field, plus the choice of notation is not consistent: this is evident when previous works are mentioned, e.g. equations 5 to 10, where the notation is the usual one, but then for the proposed approach the paper switches to the uncommon notation preferred by the author.

2) The end of the paper is very sudden. The experiments only show that, for one instance problem, the results of the proposed approach are indistinguishable from those of a less-mathematically rigorous approach, and the conclusion is that "the proposed variational formulation is well suited for rigorous derivations". But the author fails to prove that this contribution is relevant, i.e. he would need to show that his novel approach does indeed offer an advantage (preferably in terms of quality of results) with respect to ad-hoc anisotropic diffusion for some image processing problem.

Author Response

1) For the mathematical notation, I was very much in doubt on how to present the work. A detailed component notation is needed for the derivation in Equations (30)–(35), but not necessarily elsewhere. For consistency, I decided to use a component notation throughout (except for Equations (5)–(9) that deal with greyscale images). Based on your feedback, I see that it was not a good choice. In the updated manuscript, I have changed to a more conventional vector notation throughout except for the detailed derivation in Equations (30)–(35) where it is important to distinguish explicitly between colour coordinates and spatial coordinates.

2) I have added two more example applications and split the result chapter into several sections discussing the implementation details, the diffusion tensor, and the applications. Unfortunately, I cannot provide an improvement over state-of-the-art in terms of quality of the results, and that is not what I am claiming, either. In the updated paper, I have tried to emphasise that the main contribution is the variational framework as a general method to derive anisotropic gradient-domain image processing methods.

Reviewer 2 Report

The author presents a unifying variational formulation of the anisotropic gradient-domain image processing. The paper is concise and well written, and tests are presented on several images. 

There are a few typos to be removed, for example on page 3 first line, Saprio should be Sapiro.

In addition, the word "Equation" is sometimes redundant before the number of the equation and could be removed.

Finally, I think that the section "Example Application" could be enriched with the details on the implementations of the used algorithms, as well as with the comparisons on the efficiency (i.e. stopping criterion, number of iterations, ...)     

After the minor changes above, I propose to accept the manuscript.

Author Response

Based on your feedback and the comments of the other reviewer, the section on example applications has been extended significantly, and renamed to "Results and Discussion". It now has subsections on the implementation details, the diffusion tensor, and three demo applications: linear and non-linear local contrast enhancement, and colour image daltonisation.

Round 2

Reviewer 1 Report

The author has fully addressed my original concerns, in my view the manuscript can be accepted as is.